# Graph Agnostic Causal Bayesian Optimization

**Sumantrak Mukherjee**[1][*]   **Mengyan Zhang**[2][*]   **Seth Flaxman**[2]   **Sebastian Vollmer**[1,3]

[1]Department of Data Science and its Applications, DFKI GmbH
[2]Department of Computer Science, University of Oxford
[3]Department of Computer Science, University of Kaiserslautern-Landau

## Abstract

We study the problem of globally optimising a target variable of an *unknown* causal graph on which a sequence of soft or hard interventions can be performed. The problem of optimising the target variable associated with a causal graph is formalised as Causal Bayesian Optimisation (CBO). We study the CBO problem under the *cumulative regret* objective with unknown causal graphs for two settings, namely structural causal models with hard interventions and function networks with soft interventions. We propose Graph Agnostic Causal Bayesian Optimisation (GACBO), an algorithm that actively discovers the causal structure that contributes to achieving optimal rewards. GACBO seeks to balance exploiting the actions that give the best rewards against exploring the causal structures and functions. To the best of our knowledge, our work is the first to study causal Bayesian optimization with cumulative regret objectives in scenarios where the graph is unknown or partially known. We show our proposed algorithm outperforms baselines in simulated experiments and real-world applications.

## 1   Introduction

Bayesian Optimization (BO) is a robust technique for optimizing black-box functions, widely used in fields like drug discovery, robotics, and automated machine learning [Močkus, 1975, Garnett, 2023]. Traditional BO methods [Srinivas et al., 2009, Garnett, 2023] often treat functions as black boxes, but real-world data usually exhibits structural patterns. Causal Bayesian Optimization (CBO) methods [Aglietti et al., 2020, Sussex et al., 2022] leverage these structures to improve sample efficiency. However, in many cases, causal graphs are either unknown or incorrectly specified. We address this by proposing a *Graph Agnostic Causal Bayesian Optimization* (GACBO) method that works with unknown or partially known causal graphs. Unlike previous methods [Branchini et al., 2023, Alabed and Yoneki, 2022b, Toth et al., 2022], which focus on hard interventions and simple regret, have a limited prior on graphs or target causal reasoning but not BO, GACBO handles

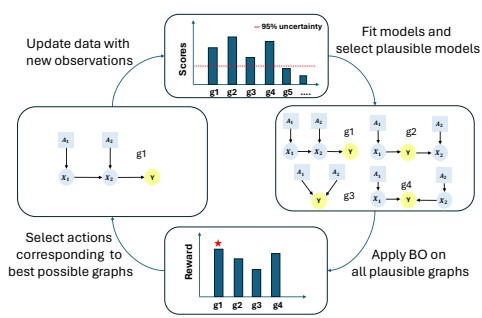

Figure 1: Graph Agnostic Causal Bayesian Optimisation (GACBO) workflow. *Top*: Select plausible graphs based on data collected so far, *Right*: Perform Causal Bayesian Optimisation on plausible graphs, *Bottom*: Select the action based on the highest reward among all plausible graphs, *Left*: Execute selected action, collect Data and repeat steps.

---

[*]Equal Contribution

Workshop on Bayesian Decision-making and Uncertainty, 38th Conference on Neural Information Processing Systems (NeurIPS 2024).

both soft and hard interventions and aims to
maximize cumulative rewards while learning the causal structure as needed. Our approach balances exploitation (selecting actions with the highest potential outcomes) and exploration (uncertainty in function space or causal structure learning). Figure 1 illustrates our proposed method. We begin with a uniform prior over all possible acyclic graph structures, modeling surrogate functions for each target node's ancestors using Gaussian processes, with inputs being the node's parents and influencing actions. The Bayesian Score [Friedman and Nachman, 2013] models the probability of these graphs. At each iteration, we retain functions and graphs within high-probability confidence intervals, selecting interventions with a UCB-based acquisition function using the reparametrization trick [Sussex et al., 2022]. Our **contributions** are as follows: **1)** We are the first to study causal Bayesian optimization with a cumulative objective in scenarios where the graph is unknown or partially known. **2)** We propose a novel algorithm, Graph Agnostic Causal Bayesian Optimization (GACBO), that handles both soft and hard interventions and includes all possible graphs in the prior, effectively sharing information across different experiments through a model-based approach. **3)** We introduce an *Upper Confidence Bound*-based acquisition function that integrates causal discovery as a subtask, engaging in it only when distinguishing between graphs improves outcomes, thereby balancing exploitation and exploration. **4)** We demonstrate on synthetic and real-world causal graphs that our algorithm performs competitively compared to existing baselines.

## 2 Problem Setup

**Structural Causal Models** An SCM [Pearl, 2009] is defined as a tuple $\langle g, Y, \boldsymbol{V}, \boldsymbol{f}_g, \boldsymbol{\Omega} \rangle$, where $g$ is a Directed Acyclic Graph (DAG) describing the relations between observed random variables $\boldsymbol{V} = \{V_i\}_{i=0}^{m-1}$, with each node $i \in [m]$ belonging to a compact space $\mathcal{V}_i \subset \mathbb{R}$. Here, $Y = V_m$ is the reward variable, and $\boldsymbol{f}_g = \{f_i^g\}_{i=0}^m$ represents the unknown functions associated with $g$, with independent noise terms $\boldsymbol{\Omega} = \{\Omega_i\}_{i=0}^m$ having zero mean and a known distribution. The parent nodes of any node $i$ in $g$ are denoted by $pa_g(i) \subset [m]$, and $\boldsymbol{Z}_i^g = \{V_j\}_{j \in pa_g(i)}$ represents the parents of node $i$ in $g$. Each node $V_i \in \boldsymbol{V}$ is generated by the function $f_i^g : \mathcal{Z}_{i,g} \to \mathcal{V}_i$, with the observed value $v_i$ given by $v_i = f_i^g(\boldsymbol{z}_i^g) + \omega_i$. These functions are evaluated in topological order from the root to the leaf nodes according to $g$. In this setting, not all observable variables are intervenable [Lee and Bareinboim, 2019]. Let $\mathcal{I} \subset \{0, \ldots, m-1\}$ denote the indices of intervenable variables. The set of observed variables $\boldsymbol{V}$ is decomposed into intervenable variables $\boldsymbol{X} = \{V_j\}_{j \in \mathcal{I}}$ and non-intervenable variables $\boldsymbol{C} = \{V_j\}_{j \notin \mathcal{I}}$, with the target variable $Y = V_m$ assumed to be non-intervenable.

**Interventions** We use a *soft intervention* model for noisy function networks (NFNs) [Eberhardt and Scheines, 2007], where controllable action variables $\boldsymbol{a} = \{a_j\}_{j=0}^n$ are added as nodes in $g$ and act as parents to nodes $V_i$, making them inputs to $f_i^g$. The subset $\boldsymbol{a}_i^g \subset \boldsymbol{a}$ affects $V_i$ based on $g$, with the action space $\mathcal{A}_i^g \subset \mathbb{R}^{|\boldsymbol{a}_i^g|}$ and total action space $\mathcal{A}$. Since $f_i^g : \mathcal{Z}_i^g \times \mathcal{A}_i^g \to \mathcal{V}_i$ is unknown, the agent cannot predict the effect of $\boldsymbol{a}_i^g$ on $V_i$ in advance. Observations are modelled as a special case i.e., $\{a_j = 0 \ \forall \ a_j \in \boldsymbol{a}_i^g\}$

*Hard interventions* are modelled as a subset of intervenable variables $\boldsymbol{I} \in \mathcal{P}(\mathcal{I})$ being set to values $\boldsymbol{a}_{\boldsymbol{I}} = \{a_i\}_{i \in \boldsymbol{I}}$ independent of their parents using the do operator s.t. $\{do(x_i = a_i) \ \forall i \in \boldsymbol{I}, a_i \in \mathcal{A}_i \subset \mathbb{R}\}$ and values are evaluated for all nodes $i \in [m]$, based on the topological ordering of $g$

$$v_i = \begin{cases} f_i^g(\boldsymbol{z}_i^g) + \omega_i & \text{if } i \notin \boldsymbol{I} \\ a_i & \text{if } i \in \boldsymbol{I} \end{cases} \tag{1}$$

**Problem Statement and Performance Metric** We address the problem of an agent interacting with an SCM or NFN defined by $\langle g^*, Y, \boldsymbol{V}, \boldsymbol{f}_{g^*}, \boldsymbol{\Omega} \rangle$, where the graph $g^*$ and functions $\boldsymbol{f}_{g^*}$ are unknown but fixed. At each round $t$, the agent selects soft intervention actions $\boldsymbol{a}_{:,t} = \{\boldsymbol{a}_{i,t}^g\}_{i=0}^m$, and for hard interventions, it chooses the nodes $\boldsymbol{I}_t$ and the corresponding intervention values $\boldsymbol{a}_{\boldsymbol{I}_t}$. The agent then collects data $\boldsymbol{v}_t$, with the subscript $t$ indicating the time step of the intervention and data collection. The objective is to design a sequence of actions $\{\boldsymbol{a}_{:,t}\}_{t=0}^T$ or $\{\boldsymbol{I}_t, \boldsymbol{a}_{\boldsymbol{I}_t}\}_{t=0}^T$ that maximizes the average expected reward for soft and hard interventions, respectively, which is equivalent to minimizing the expected cumulative regret [Sussex et al., 2022, Lattimore and Szepesvári, 2020]:

$$R_T = \sum_{t=1}^T \left[\mathbb{E}[y|\boldsymbol{a}^*] - \mathbb{E}[y|\boldsymbol{a}_{:,t}]\right]; \quad R_T = \sum_{t=1}^T \left[\mathbb{E}[y|\boldsymbol{a}^*] - \mathbb{E}[y|do(\boldsymbol{x}_{\boldsymbol{I}_t} = \boldsymbol{a}_{\boldsymbol{I}_t})]\right]. \tag{2}$$

---

**Algorithm 1** Graph Agnostic Causal Bayesian Optimisation(Soft Interventions) (GACBO-S)

---

**Input:** Parameters $\{\beta_t\}_{t \geq 1}$, $\Omega$, generic kernel function $k_i$, prior over possible $\psi_{i,0}$ graph components, prior means $\mu_{i,0} = 0 \ \forall \ i \in [m]$.

**for** $t = 1 \ldots T$ **do**

    Construct confidence bounds for plausible functions $\mathcal{M}_t$ as in Eq. (4).

    Construct plausible graphs $G_t$ as in Eq. (5) using Algorithm 3.

    Select $\boldsymbol{a}_{:,t} \in \arg \max_{\boldsymbol{a} \in \mathcal{A}} \max_{g \in G_t} \max_{\boldsymbol{\eta}_g(\cdot)} \mathbb{E}[y | \tilde{\boldsymbol{f}}_g, \boldsymbol{a}]$ as in Eq. (8).

    Observe all nodes $\boldsymbol{v}_t$ and update $\mathcal{D}_t = \mathcal{D}_{t-1} \cup \{\boldsymbol{v}_t, \boldsymbol{a}_{:,t}\}$.

    Update posterior $\{\{\mu_{i,t}^g(\cdot), \sigma_{i,t}^g(\cdot)\}_{i=0}^m\}_{g \in G}$.

**end for**

---

## 3 Method

In this section, we propose the *Graph Agnostic Causal Bayesian Optimisation (*GACBO*)* and outline the technical details we utilise to tackle the different sources of uncertainty: **1) Function Uncertainty**: For each node $i$ in graph $g$ with parent set $\boldsymbol{z}_i^g$, we model the functional relationship using Gaussian Processes (GPs) as per standard Bayesian Optimization (BO) practices [Garnett, 2023]. Let $\mu_{i,0}^g$ and $(\sigma_{i,0}^g)^2$ denote the prior mean and variance of the function $f_{i,g}^t$ for all $i \in [m]$ and $g \in G_0$. These are updated at time step $t$ based on previous data $\mathcal{D}_t = \{(\boldsymbol{z}_{i,1}^g, \boldsymbol{a}_{i,1}^g), \ldots, (\boldsymbol{z}_{i,t-1}^g, \boldsymbol{a}_{i,t-1}^g)\}$ with standard GP update formula A.3. Kernel choices align with our regularity assumptions A.6. **2) Process Uncertainty**: We account for additive noise in data generation, assuming it to be either bounded or sub-Gaussian, ensuring function domains remain compact. Sub-Gaussian noise $\omega_i \sim \Omega_i \ \forall i \in [m]$ is incorporated into our GP variance terms $\rho_i^2$. **3) Causal Graph Structure Uncertainty**: Assuming no prior knowledge of the DAG structure, we treat all DAGS as equally likely. Using the Markov Property of Bayesian Networks, we decompose the graph into components (parent sets of observable nodes) and model their likelihood given data $\mathcal{D}_t$, with a uniform prior over all possible parent sets. The likelihood is computed using the *Score* [Friedman and Nachman, 2013], detailed in Section A.4.

**Plausible Models** At time step $t$, plausible models are defined as the set of surrogate models likely to contain the true SCM, with confidence intervals ensuring a probability of at least $1 - \delta$. We calculate the plausible functions for each node $i$ using:

$$|\tilde{f}_{i,t}^g(\boldsymbol{z}_i^g, \boldsymbol{a}_i^g) - \mathbb{E}_{i,t}[\boldsymbol{z}_i, \boldsymbol{a}_i]| \leq \beta_{i,t} \sqrt{\mathbb{V}_{i,t}[\boldsymbol{z}_i, \boldsymbol{a}_i]}, \tag{3}$$

$$\mathbb{E}_{i,t}[\boldsymbol{z}_i, \boldsymbol{a}_i] = \mathbb{E}_{g \sim p(g|\mathcal{D}_t)}[\mu_{i,t-1}^g(\boldsymbol{z}_i^g, \boldsymbol{a}_i^g)],$$

$$\mathbb{V}_{i,t}[\boldsymbol{z}_i, \boldsymbol{a}_i] = \mathbb{V}_{g \sim p(g|\mathcal{D}_t)}[\mu_{i,t-1}^g(\boldsymbol{z}_i^g, \boldsymbol{a}_i^g)] + \mathbb{E}_{g \sim p(g|\mathcal{D}_t)}[(\sigma_{i,t-1}^g(\boldsymbol{z}_i^g, \boldsymbol{a}_i^g))^2].$$

Here, $\boldsymbol{z}_i$ includes all observable nodes except $V_i$, and $\beta_{i,t}$ ensures confidence bounds, set as $\beta_{i,t} = \beta_T$. Further details about $\beta_{i,t}$ are provided in A.6. At time $t$, all $\tilde{f}_i^g$ within the confidence intervals defined by the joint posterior $p(g|\mathcal{D}_t)$ and associated GP posteriors form the *plausible functions* $\mathcal{M}_t$:

$$\mathcal{M}_t^g = \{\tilde{\boldsymbol{f}}_g = \{\tilde{f}_i^g\}_{i=0}^{|V|} \text{ such that } \forall \ i : \tilde{f}_i^g \in \mathcal{H}_{k_i}, \|\tilde{f}_i^g\|_{k_i} \leq \mathcal{B}_i, \text{ and (3) holds for all } a \in \mathcal{A}\}. \tag{4}$$

The *plausible graphs* $G_t$ are defined as:

$$G_t = \{g \mid \forall_i \exists \tilde{f}_i^g \in \mathcal{M}_t^g, g \in G_{t-1}\}, \tag{5}$$

with $G_0 = \{g \in \mathcal{G}\}$, where $\mathcal{G}$ denotes the space of all possible DAGs. For a graph to be plausible at time $t$, there must be at least one associated function within the confidence interval for each node.

**Algorithm** We present two variants of the GACBO algorithm for soft 1 and hard interventions 2. The causal subgraph discovery algorithm 3 is used in both variants of the algorithm to estimate the graph posteriors and sample relevant graphs.

The maximum possible value varies across different graphs and function sets within the plausible models at time $t$, each having its own optimal action. Our acquisition function identifies the graph and function set that yield the highest target node value and returns the action that achieves it.

$$\boldsymbol{a}_{:,t} = \arg \max_{\boldsymbol{a} \in \mathcal{A}} \max_{g \in G_t} \max_{\tilde{\boldsymbol{f}}_g \in \mathcal{M}_t^g} \mathbb{E}[y | \tilde{\boldsymbol{f}}_g, \boldsymbol{a}]. \tag{6}$$

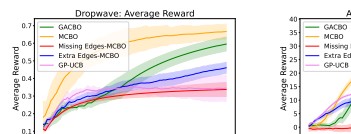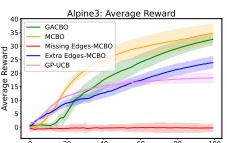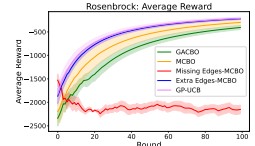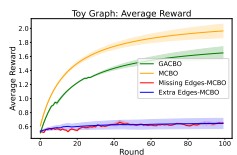

Figure 2: Simulation results comparing GACBO with MCBO (true and incorrect graphs) and GP-UCB. GP-UCB is not applicable to ToyGraph due to hard interventions.

It is important to note that 6 is not suitable for standard optimization methods because it requires maximization over a set of functions with a bounded RKHS norm.

Therefore for a graph $g$ within the plausible models, the reparametrisation trick introduced in Curi et al. [2020] and utilised for CBO in Sussex et al. [2022] can be used to write any function $\tilde{f}_i^g \in \tilde{f}_g \in \mathcal{M}_t^g$ using $\eta_{i,g} : \mathcal{Z}_i^g \times \mathcal{A}_i^g \to [-1,1]$, as

$$\tilde{f}_{i,t}^g(\tilde{z}_i^g, \tilde{a}_i^g) = \mu_{i,t-1}^g(\tilde{z}_i^g, \tilde{a}_i^g) + \beta_t \sigma_{i,t-1}^g(\tilde{z}_i^g, \tilde{a}_i^g)\eta_{i,g}(\tilde{z}_i^g, \tilde{a}_i^g), \tag{7}$$

The acquisition function can therefore be expressed in terms of $\boldsymbol{\eta_g} : \mathcal{Z}^g \times \mathcal{A}^g \to [-1,1]^{|V(g)|}$, where $|V(g)|$ is the number of nodes in the graph $g$,

$$\arg\max_{\boldsymbol{a}\in\mathcal{A}} \max_{g\in G_t} \max_{\boldsymbol{\eta_g}(\cdot)} \mathbb{E}[y|\tilde{\boldsymbol{f}}_g, \boldsymbol{a}]. \tag{8}$$

More details about the optimistic reparameterisation trick can be found in A.5. The data collected is used to update model posteriors and construct plausible models for next time step.

## 4   Results

We evaluate GACBO on synthetic environments (Dropwave, Alpine3, Rosenbrock, ToyGraph) and a real-world Epidemiology Graph from [Astudillo and Frazier, 2021, Branchini et al., 2023]. The metric used is the average reward, inversely related to cumulative regret. We repeat each experiment 5 times with different seeds and report average rewards $\pm\sigma/\sqrt{5}$, where $\sigma$ is the standard deviation. We compare GACBO with: 1) MCBO [Sussex et al., 2022] using the true causal graph, 2) MCBO with incorrect graphs (missing or extra edges), 3) GP-UCB [Srinivas et al., 2009] for soft interventions.

**Simulations**   Figure 2 shows our results. For soft interventions, we use Dropwave, Alpine3, and Rosenbrock. For hard interventions, we use ToyGraph [Aglietti et al., 2020]. MCBO with the true graph generally performs best, except in Rosenbrock, where GP-UCB excels due to the function's additive structure. MCBO struggles with incorrect graphs, particularly when extra edges increase dimensionality or missing edges misrepresent the function space. GACBO, initially hampered by lack of graph information, quickly learns the correct structure and matches MCBO's performance after about 100 rounds. Further information regarding performance in specific environments can be found in A.9.

**Real-World Application: Epidemiology**   We test GACBO in an Epidemiology setting [Havercroft and Didelez, 2012, Branchini et al., 2023], aiming to minimize HIV viral load by choosing from interventions $\mathcal{I} = \{\emptyset, \{T\}, \{R\}, \{T, R\}\}$. Despite the environment's complexity, GACBO quickly learns the correct causal structure and matches MCBO's performance with the true graph within 100 rounds, significantly outperforming MCBO with incorrect graphs.

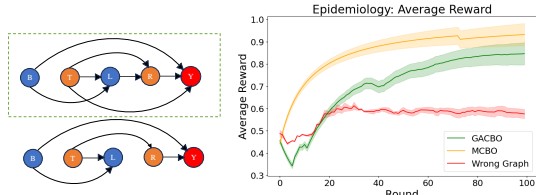

Figure 3: Epidemiology application. Top left: true causal graph. Bottom left: incorrect causal structure for MCBO. Right: performance comparison.

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

# A APPENDIX

## A.1 Nomenclature

Using standard notation, we use Capital letters to denote random variables and lowercase letters to denote the realization of said random variables. We use bold letters to denote sets of certain nodes. The support of a variable is given by curly letters. We use the subscript $t$ to index data observed thus far, and the subscript $i$ is used to index a particular node in a vector, the superscript $g$ is used to refer to the input space

| Symbol | Description |
|--------|-------------|
| $V_j$ | $j^{\text{th}}$ observed variable |
| $Y$ | Target variable we seek to optimize corresponds to $V_m$ |
| $\boldsymbol{V}$ | Set of all observed variables |
| $\boldsymbol{X}$ | Set of intervenable variables |
| $\boldsymbol{C}$ | Set of non intervenable variables |
| $A_i$ | Action performed on node i |
| $g^*$ | True latent causal graph |
| $\boldsymbol{A}$ | Action vector composed of $\{A_i\}_{i=0}^m$ |
| $\boldsymbol{Z}_i^{g^*}$ | The parents of node $i$ in graph $g^*$ |
| $f_i^{g^*}(\boldsymbol{z}_i^{g^*}, \boldsymbol{a}_i^{g^*})$ | functions relating a node $i$ with its parents and actions |
| $\boldsymbol{f}^{g^*}(\boldsymbol{a})$ | The overall function with input action composed of functions $\{f_i^{g^*}\}_{i=0}^m$ related by graph $g^*$ |
| $\boldsymbol{F}_{g^*}$ | the set of respective unknown functions associated with $g^*$, i.e. $\{f_i^{g^*}\}_{i=0}^m$ |
| $\boldsymbol{\Omega}$ | a set of independent noises with zero mean and known distribution, i.e. $\{\Omega_i\}_{i=0}^m$ |
| $pa_{g^*}(i)$ | indices of parent nodes of any node, defined for the DAG $g^*$ |
| $\{y_t, \boldsymbol{v}_t, \boldsymbol{a}_t\}$ | Observation of reward variable $y_t$ and intermediate variables $v_t$ for the corresponding action $\boldsymbol{a}_t$ |
| $\mathcal{D}_t$ | Observations for actions until time $t$, is the set $\{y_j, \boldsymbol{v}_j, \boldsymbol{a}_j\}_{j=0}^t$ |
| $G_t$ | Posterior of the distribution over graphs at time $t$ |
| $g$ | Random DAG samples from $G_t$ |
| $k_i^g(\cdot, \cdot)$ | Kernel defined on input space implied by graph $g$ for node $i$, gives covariance between two points |
| $\boldsymbol{k}_{i,t}^g(\cdot)$ | A vector of covariances of the current input to previous inputs $[k_i^g((z_{i,t}^g, a_{i,t}^g), \cdot)]_{i=0}^t$ |
| $\boldsymbol{K}_i^g$ | Covariance matrix based on previous $\mathcal{D}_t$ |
| $\mu_{i,t}^g(\cdot)$ | Mean function based on data $\mathcal{D}_t$ and kernel $k_i^g(\cdot, \cdot)$ |
| $\sigma_{i,t}^g(\cdot)$ | Variance function based on data $\mathcal{D}_t$ and kernel $k_i^g(\cdot, \cdot)$ |
| $GP(\mu_{i,t}^g, \sigma_{i,t}^g)$ | Gaussian Process $\tilde{f}_i^g(\cdot) \sim \mathcal{N}(\mu_{i,t}^g(\cdot), \sigma_{i,t}^g(\cdot))$ |
| $\mathcal{H}_{k_i^g}$ | Hilbert Spaces of functions implied by kernel $k_i^g$ |
| $\tilde{f}_i^g$ | A function sampled from Gaussian Processes GP |
| $\omega_i$ | Observational noise of node $i$ |
| $\mathcal{M}_t$ | Plausible models at time $t$ based on confidence bounds |

## A.2 Related Work

**Causal Decision Making** The first causal Bayesian optimisation setting was proposed in Aglietti et al. [2020], which focused on hard interventions and the best intervention identification setting.

Sussex et al. [2022] expanded their setting to include soft interventions and noisy environments. They proposed the Model-based Causal Bayesian Optimisation (MCBO) algorithm, which is the state-of-the-art method with a known graph. With unknown graphs for cumulative regret objective, Lu et al. [2021], De Kroon et al. [2022], Konobeev et al. [2023] considered causal multi-armed bandits. Lu et al. [2021] studied causal trees, causal forests and proper interval graphs, with regret analysis under a few causal assumptions. De Kroon et al. [2022] utilised an estimator based on separating sets, with no theoretical analysis on regret shown. Konobeev et al. [2023] proposed a RAndomized Parent Search algorithm (RAPS) and showed conditional regret upper bounds. Malek et al. [2023] show that the unknown causal graph be exponentially hard in parents of the outcome and studies the problem under the additive assumption on the outcome. All the above work considered discrete arms (intervention values) and linear bandits, while our work addresses continuous intervention values, non-linear relations between nodes and a more general class of graphs.

Branchini et al. [2023] studied the CBO setting with an unknown graph for the best intervention identification setting. Their approaches are based on the entropy search criterion. However, directly applying their method to cumulative regret objective would lead to suboptimal performance since one needs to further balance the exploitation-exploration balance between picking actions that lead to the best rewards and learning causal structures. Alabed and Yoneki [2022a] studied the CBO problem for unknown causal graph scenarios with a specific application to autotuners. To the best of our knowledge, we are the first to study the CBO with unknown graph and cumulative regret objectives.

**Active Causal Discovery** von Kügelgen et al. [2019] developed a Bayesian optimal experimental design framework to perform active causal discovery for Gaussian Process networks. Lorch et al. [2021], Giudice et al. [2023] addressed the problem of causal discovery for graphs with a larger number of nodes. Based on this, Tigas et al. [2022, 2023] performed active causal discovery for larger graphs. Toth et al. [2022] considered the active learning methods for unifying sequential causal discovery and causal reasoning.

The goals of active causal discovery and Bayesian optimisation are misaligned. While Bayesian optimisation tries to balance exploration and exploitation to minimise cumulative regret, the active causal discovery acquisition function might choose an intervention that has a low reward and does not help future steps of CBO but helps discover the true underlying Causal Graph. Therefore it is sub-optimal to first perform active causal discovery and then followed by causal Bayesian optimisation as separate steps. Our algorithm naturally unifies these two steps by making causal discovery a sub-task of causal Bayesian optimisation. See Appendix A.10 for a detailed discussion.

## A.3 Surrogate models

Surrogate models help us incorporate our prior beliefs into the modelling process and allow us to enact interventions without performing them in the real environment and also quantify the total uncertainty related to certain outcomes. Define surrogate model $m_t \sim \mathcal{M}_t$ at time step $t$ as a triple $m_t = (g_t, \tilde{\boldsymbol{f}}_t^g, \boldsymbol{\omega}_t^2)$. $\mathcal{M}_t$ denotes the posterior of plausible models, $g_t \sim G_t$ is one possible realisation of posterior $G_t$ at time $t$, $\tilde{\boldsymbol{f}}_t^g = \{\tilde{f}_{t,i}^g\}_{i=0}^m$ where the surrogate function $\tilde{f}_{t,i}^g \in \mathcal{H}_{k_i^g}$ belongs to the RKHS $\mathcal{H}_{k_i^g}$ which is defined on the input space $\mathcal{S}_i^g = \mathcal{Z}_i^g \times \mathcal{A}_i^g$ (and $\mathcal{S}_i^g = \mathcal{Z}_i^g$ for hard interventions) for all nodes $i$ as implied by kernel $k_i^g : \mathcal{S}_i^g \times \mathcal{S}_i^g \to \mathbb{R}$. And we assume subgaussian observational noise of each node $\boldsymbol{\omega}_t^2 = \{\omega_{t,i}^2\}_{i=0}^m$.

**Surrogate Functions** We model surrogate functions using Gaussian processes (GPs). Posterior means $\mu_{i,t}^g$ and variances $\sigma_{i,t}^g$ for any function parameterising any possible graph $g$ at a given point is calculated according to the GP posterior at time step $t$. The posterior is calculated using GP update equations [Williams and Rasmussen, 1995].

$$\tilde{f}_{i,t}^g(\boldsymbol{z}_i^g, \boldsymbol{a}_i^g) \sim \mathcal{N}(\mu_{i,t}^g(\boldsymbol{z}_i^g, \boldsymbol{a}_i^g), \sigma_{i,t}^g(\boldsymbol{z}_i^g, \boldsymbol{a}_i^g)), \tag{9}$$

where

$$\mu_{i,t}^g(\boldsymbol{z}_i^g, \boldsymbol{a}_i^g) = \boldsymbol{k}_{i,t}^g(\boldsymbol{z}_i^g, \boldsymbol{a}_i^g)^\top(\boldsymbol{K}_i^g + \rho_i^2\boldsymbol{I})^{-1}\text{vec}(v_{i,1:t});$$
$$\sigma_{i,t}^g(\boldsymbol{z}_i^g, \boldsymbol{a}_i^g) = k_i^g((\boldsymbol{z}_i^g, \boldsymbol{a}_i^g), (\boldsymbol{z}_i^g, \boldsymbol{a}_i^g)) - \boldsymbol{k}_{i,t}^g(\boldsymbol{z}_i^g, \boldsymbol{a}_i^g)^\top(\boldsymbol{K}_i^g + \rho_i^2\boldsymbol{I})^{-1}\boldsymbol{k}_{i,t}^g(\boldsymbol{z}_i^g, \boldsymbol{a}_i^g), \tag{10}$$

where $\boldsymbol{I}$ is used to define the identity matrix, $\text{vec}(v_{i,1:t}) = [v_{i,1} \ldots v_{i,t}]^\top$, $\boldsymbol{k}_{i,t}^g(\boldsymbol{z}_i^g, \boldsymbol{a}_i^g) = [k_i^g((\boldsymbol{z}_{i,1}^g, \boldsymbol{a}_{i,1}^g), (\boldsymbol{z}_i^g, \boldsymbol{a}_i^g)), \ldots, k_i^g((\boldsymbol{z}_{i,t}^g, \boldsymbol{a}_{i,t}^g), (\boldsymbol{z}_i^g, \boldsymbol{a}_i^g))]$, $[\boldsymbol{K}_i^g]_{t_1,t_2} = k_i^g((\boldsymbol{z}_{i,t_1}^g, \boldsymbol{a}_{i,t_1}^g), (\boldsymbol{z}_{i,t_2}^g, \boldsymbol{a}_{i,t_2}^g))$.

**Graph Likelihood** The *Markov Property* of Bayesian networks allows for a compact factorisation of the joint distribution of all observed nodes $\boldsymbol{V} = \{V_1, \ldots, V_m\}$ in the Bayesian Network,

$$p(\boldsymbol{V}|g) = \prod_{i=0}^m p(V_i|\boldsymbol{Z}_i^g). \tag{11}$$

The joint distribution factorises into conditional distributions given its parents in the graph $g$.

In the case of soft interventions any observed node $V_i$ is affected by its parents $\boldsymbol{Z}_i^g$ as well as the actions which appear as extra nodes in the SCM, we use $\boldsymbol{A}_i^g$ to denote the set of action nodes affecting node $i$ therefore it is calculated as

$$p(\boldsymbol{V}|g) = \prod_{i=0}^m p(V_i|\boldsymbol{Z}_i^g, \boldsymbol{A}_i^g) \tag{12}$$

The distribution factorises into conditional distributions for each variable, given its parents in the DAG and the associated actions for the node.

GPs admit a closed-form expression for the marginal likelihood of the $t$ observations $v_{i,1:t}$ of the node $V_i$. $p(v_{i,1:t}|g, \boldsymbol{\theta}_i)$ can be calculated as below

$$(2\pi)^{-\frac{t}{2}} |\tilde{\boldsymbol{K}}_{i,\theta}^g|^{-\frac{1}{2}} \exp\left(-\frac{1}{2}v_{i,1:t}^\top(\tilde{\boldsymbol{K}}_{i,\theta}^g)^{-1}v_{i,1:t}\right) \tag{13}$$

where $\tilde{\boldsymbol{K}}_{i,\theta}^g = \boldsymbol{K}_{i,\theta}^g + \omega_i^2 I$. The covariance matrix $\boldsymbol{K}_{i,\theta}^g$ is given by the kernel $k_{i,\theta}^g$ used and observations collected until time step $t$ $(\boldsymbol{z}_{i,1}^g, \boldsymbol{a}_{i,1}^g) \ldots (\boldsymbol{z}_{i,t}^g, \boldsymbol{a}_{i,t}^g), (\boldsymbol{z}_{i,1}^g) \ldots (\boldsymbol{z}_{i,t}^g)$ for soft and hard interventions respectively. The input space of the functions and hence the kernel specified is dependent on the selected graph. The lengthscales $\boldsymbol{\theta}_i = \{\theta_{i,j}\}_{i \in pa_g(i)}$ chosen for different input nodes in the selected graph, determine the smoothness of the functions in the RKHS implied by the kernel. The lengthscales chosen for the kernel relate directly to the smoothness of the functions sampled from the GP [Berkenkamp et al., 2019]. We define priors $\boldsymbol{\theta}_i \sim \pi(\boldsymbol{\theta}_i)$ over hyperparameters consistent with our smoothness assumptions.

## A.4  Bayesian Score

The *Score* is defined as Friedman and Nachman [2013] as $S$ and is calculated as follows. The score shows the probability of the observed values of node $V_i$ is $v_{i,1:t}$ given the graph $g$ and dataset $\mathcal{D}_{t-1}$, where graph $g$ indicates the parents of node $V_i$ is $\boldsymbol{Z}_i^g$ and actions $\boldsymbol{A}_i^g$.

$$S(V_i, \boldsymbol{Z}_i^g, \boldsymbol{A}_i^g|\mathcal{D}_t) = \int p(v_{i,1:t}|g, \boldsymbol{\theta}_i)\pi(\boldsymbol{\theta}_i|g)d\boldsymbol{\theta}_i, \tag{14}$$

Therefore the probability of observing data $\mathcal{D}_t$ given $g$ is given as the product of observing the values of each node in $i \in [m]$ given the values of its parents according to graph $g$

$$P(\mathcal{D}_t|g) = \prod_{i=0}^m S(V_i, \boldsymbol{Z}_i^g, \boldsymbol{A}_i^g|\mathcal{D}_t). \tag{15}$$

The probability of the graph $g$ given $\mathcal{D}_t$ is directly proportional to the product of the probability of observing the data given graph $P(\mathcal{D}_t|g)$ and prior probability of graph $g$ $p(g)$ using Bayes Rule,

$$P(g|\mathcal{D}_t) \propto P(\mathcal{D}_t|g)p(g). \tag{16}$$

## A.5 Optimistic Reparameterisation

Actions are evaluated based on the topological ordering of nodes in a graph $g$. Finding the optimal value of the target node requires selecting the correct actions for all its ancestor nodes. This task is challenging due to two key factors. First, there is function uncertainty, meaning that for a given set of node values and a fixed action, the exact value of the affected node is not known but lies within a confidence interval. Second, the values of ancestor nodes, given actions up to node $i$ in $g$, can vary. The optimistic reparametrization approach involves choosing values for these ancestor nodes within their confidence intervals in a way that allows descendant nodes to take on values that optimize the target node.

For a graph $g$ within the set of plausible models, the reparametrization trick from Curi et al. [2020] and used for CBO in Sussex et al. [2022] allows us to express any function $\tilde{f}_i^g \in \tilde{\boldsymbol{f}}_g \in \mathcal{M}_t^g$ using $\eta_{i,g} : \mathcal{Z}_i^g \times \mathcal{A}_i^g \to [-1,1]$ as

$$\tilde{f}_{i,t}^g(\tilde{\boldsymbol{z}}_i^g, \tilde{\boldsymbol{a}}_i^g) = \mu_{i,t-1}^g(\tilde{\boldsymbol{z}}_i^g, \tilde{\boldsymbol{a}}_i^g) + \beta_t \sigma_{i,t-1}^g(\tilde{\boldsymbol{z}}_i^g, \tilde{\boldsymbol{a}}_i^g) \eta_{i,g}(\tilde{\boldsymbol{z}}_i^g, \tilde{\boldsymbol{a}}_i^g).$$

The acquisition function can then be expressed in terms of $\boldsymbol{\eta}_g : \mathcal{Z}^g \times \mathcal{A}^g \to [-1,1]^{|V(g)|}$, where $|V(g)|$ is the number of nodes in graph $g$:

$$\arg \max_{\boldsymbol{a} \in \mathcal{A}} \max_{g \in G_t} \max_{\boldsymbol{\eta}_g(\cdot)} \mathbb{E}[y | \tilde{\boldsymbol{f}}_g, \boldsymbol{a}].$$

By selecting a sequence $\boldsymbol{\eta}_g$ corresponding to actions $\boldsymbol{a}_{:,t}$, we can choose optimistic but plausible functions that maximize the target node for a given action. The objective then becomes finding the right sequence $\boldsymbol{\eta}_g$ and the optimal action $\boldsymbol{a}_{:,t}$ to achieve the best possible outcome given the current function estimates.

In noiseless settings, instead of parameterizing each $\eta_i$ as a neural network, it can be parameterized as a constant. Without noise, the inputs to $\eta_{i,g}$ ($z_i^g$ and $a_i^g$) are fixed given $\boldsymbol{a}$, reducing the parameter space to optimize. This allows us to use the optimization procedure from EIFN by Astudillo and Frazier [2021], which is also implemented as an optimizer in the BoTorch package Balandat et al. [2020].

For noisy settings where each $\eta_{i,g} : \mathcal{Z}_i^g \times \mathcal{A}_i^g \to \mathbb{R}$ is parameterized as a neural network, we use our own optimizer. For each initialization of $\eta$ parameters, we perform stochastic gradient descent to optimize both the $\eta_i$ parameters and the action $a$, leveraging the differentiability of the acquisition function with respect to both $a$ and the $\eta_i$ parameters. After running stochastic gradient descent on many random initializations, we select the candidate with the highest acquisition function value. Since the acquisition function may be highly non-convex, we use a large number of random initializations. Alternative approaches, such as those in Curi et al. [2020] for model-based reinforcement learning, could also be adapted to optimize our acquisition function. When parameterizing each $\eta_i$ with a neural network, we use a two-layer feed-forward network with a ReLU non-linearity, followed by an element-wise Sigmoid function to map the output into $[-1,1]$.

## A.6 Regularity Assumption

We operate under standard smoothness assumptions for any function relating any node to its parents $f_i^{g^*} \to \mathcal{S} \times \mathcal{V}_i$ is defined over a compact domain $\mathcal{S}$. For all nodes $i \in [m]$, we assume $f_i^{g^*}(\cdot)$ belongs to a reproducible Kernel Hilbert Space (RKHS) $\mathcal{H}_{k_i^{g^*}}$, a space of smooth functions defined on the input space $\mathcal{S} = \mathcal{Z}_i^{g^*} \times \mathcal{A}_i^{g^*}$ for FNs and $\mathcal{S} = \mathcal{Z}_i^{g^*}$ for SCMs. This means all functions $f_i^{g^*} \in \mathcal{H}_{k_i^{g^*}}$ are induced by $k_i^{g^*} : \mathcal{S} \times \mathcal{S} \to \mathbb{R}$. We also assume that $k_i^{g^*}(s,s') \leq 1$ for every $s,s' \in \mathcal{S}$. We enforce our smoothness assumptions by placing a bound on the RKHS norm of $f_i^{g^*}(\cdot)$, $\|f_i^{g^*}\| \leq \mathcal{B}_i$ for some fixed constant $\mathcal{B}_i \geq 0$. To ensure the compactness of the domain $\mathcal{Z}_i^{g^*}$ we assume that the noise $\omega_i$ is either subgaussian or bounded i.e $\omega_i \in [-1,1]$.

## A.7 Supplementary Algorithms

**GACBO Hard** Similar to the acquisition function defined in 6, we define an acquisition function for hard interventions, with the only difference being that hard interventions are performed instead of

---

**Algorithm 2** Graph Agnostic Causal Bayesian Optimisation (Hard intervention) (GACBO-H)

---

**Requires:** Parameters $\{\beta_t\}_{t\geq 1}$, $\Omega$, generic kernel function $k_i$, prior over possible graphs $G_0$, prior means $\mu_{i,0}^g = 0 \forall i \in [m], g \in G_0$.

**for** $t = 1 \dots T$ **do**

    Construct confidence bounds for plausible functions $\mathcal{M}_t$ as in 4

    Construct plausible graphs as in 5.

    Select $I, \boldsymbol{a}_I \in \arg\max_{I, \boldsymbol{a}_I \in \mathcal{A}} \max_{g \in G_t} \max_{\boldsymbol{\eta}_g(\cdot)} \mathbb{E}.[y|\tilde{\boldsymbol{f}}_g, do(V_I = \boldsymbol{a}_I)]$ as in 17.

    Observe all nodes $\boldsymbol{v}_t$ and update $\mathcal{D}_t = \mathcal{D}_{t-1} \cup \{\boldsymbol{v}_t, \boldsymbol{a}_t\}$ Update posterior $\{\{\mu_{i,t}^g(\cdot), \sigma_{i,t}^g(\cdot)\}_{i=0}^m\}_{g \in G}$.

**end for**

---

---

**Algorithm 3** Causal Subgraph Discovery

---

**Input:** $\boldsymbol{S}_i = \{S(V_i, Z_i, A_i|\mathcal{D}_t), \forall (Z_i, A_i) \in \mathcal{Z}_i \times \mathcal{A}_i\}$, where $i \in [m], g_t = \{\}, De(m) = \{\}$.

**function** FINDSUBGRAPH$(g, i, \boldsymbol{S}_i, De(i))$

    **if** $i \in g$ **then**

        **return** $g$

    **end if**

    $\boldsymbol{S}_i^c = \{S \in \boldsymbol{S}_i \mid Z_i \cap De(i) = \emptyset\}, \tilde{\boldsymbol{S}}_i^c = \{\frac{S}{\sum_{S \in \boldsymbol{S}_i^c} S} \forall S \in \boldsymbol{S}_i^c\}, (Z_i^c, A_i^c) \sim \text{Multinomial}(\tilde{\boldsymbol{S}}_i^c)$

    **if** $Z_i^c = \emptyset$ **then**

        $g = g \cup \{i : (Z_i^c, A_i^c)\}$

    **else**

        $g = \{g \cup \{i : (Z_i^c, A_i^c)\}, Pa_g(i) \sim \text{Uniform}(\text{Permutations}(Z_i^c))$

        **for** $j \in Pa_g(i)$ **do**

            $De(j) = De(j) \cup \{i\}, g = \text{FINDSUBGRAPH}(g, j, \boldsymbol{S}_j, De(j))$

        **end for**

    **end if**

    **return** g

**end function**

**Output:** $g_t = \text{FINDSUBGRAPH}(g_t, m, \boldsymbol{S}_m, De(m))$

---

soft interventions. The observational uncertainty is propagated through the resulting mutilated graph, the reparameterisation trick is used to find optimistic upper confidence for all plausible graphs

$$\arg\max_{I, \boldsymbol{a}_I \in \mathcal{A}} \max_{g \in G_t} \max_{\boldsymbol{\eta}_g(\cdot)} \mathbb{E}[y|\tilde{\boldsymbol{f}}_g, do(V_I = \boldsymbol{a}_I)]. \tag{17}$$

This is slightly different as compared to soft interventions, because a hard intervention mutates the graph, making the node independent of all ancestor nodes and interventions performed on them, thus simplifying the problem. This induces the notion of Minimal Intervention Sets MIS [Lee and Bareinboim, 2018]. A MIS for an SCM $\langle g, Y, \boldsymbol{V}, \boldsymbol{f}_g, \boldsymbol{\Omega} \rangle$ is defined as the set of variables $\mathbf{X}_{\mathbf{s}} \in \mathcal{P}(\mathbf{X})$ such that there exists no such $\mathbf{X}_{\mathbf{s}}^{'} \subset \mathbf{X}_{\mathbf{s}}$ for which $\mathbb{E}[Y \mid do(\mathbf{X}_{\mathbf{s}})] = \mathbb{E}[\mathbf{Y} \mid \mathbf{do}(\mathbf{X}_{\mathbf{s}}')]$. We denote the MIS for graph $g$ with target node $y$ as $\mathbb{M}_{g,y}$ however since the graph structure is not known to us a priori, we construct our *Plausible MIS* $\mathbb{M}_{y,t}$, by taking the union over the MIS of plausible graphs at time step $t$, i.e. $\mathbb{M}_{y,t} = \bigcup_{g \in G_t} \mathbb{M}_{g,y}$. For each plausible graph, we only compare interventions within the MIS of the given graph to find the intervention which maximises the surrogate model associated with that particular graph and then compare across all possible graphs to find the best plausible intervention.

**Causal Subgraph Discovery**    Our causal Subgraph discovery algorithm is used in both versions of the GACBO algorithm to sample from the posterior over graphs. The Graph discovery algorithm only focuses on components that are possibly ancestors of the target variables. It recursively samples graph components based on scores of the graph components for any given node. The given nodes are those of already sampled components closer to the target variable. We ensure acyclicity by excluding components that form a cycle with one or more of the already sampled nodes. We randomly sample the ordering of the parents of a node to determine their respective graph components.

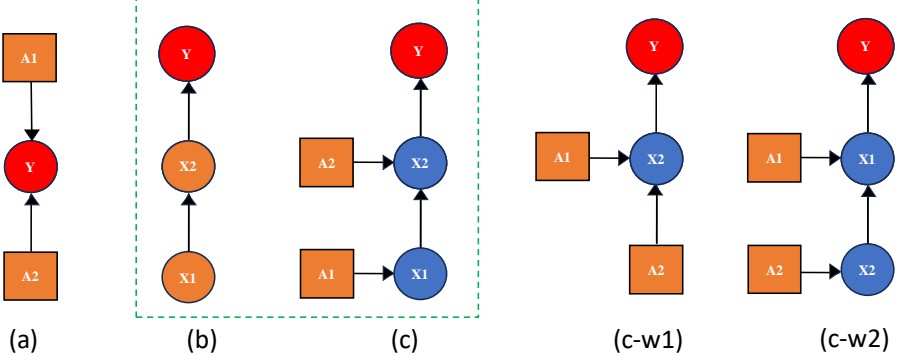

Figure 4: Problem Settings and Causal Structures: (a) Bayesian optimisation; (b) Structural causal models and hard interventions; (c) Function networks and soft interventions; (c-w1) Incomplete graph for (c), missing X1; (c-w2) Incorrect graph for (c), reversing the order of X1 and X2. The blue circles $X1$ and $X2$ represent non-manipulative variables, the orange squares $A1$ and $A2$ represent actions that can be taken, and $Y$ is the outcome of interest.

## A.8 Motivating Examples

We illustrate an example of how making use of causal structure in soft or hard intervention cases can improve the optimization in Figure 5. This example shows observing intermediate nodes is essential. Consider a modeller trying to optimize $Y$ using $A2$, while ignoring the values of $X1$, $X2$, they would observe a lot of different values of $y$ for the same action $A2$ considering that $X1$ can take on several different values, consequently affecting $X2$ and $Y$.

Consider an investigator trying to optimize her crop yield, the crop could be dependent on several variables such as hours of daylight, soil moisture content, soil nitrogen content, temperature, rainfall, pest control, fertilizer use, crop rotation, irrigation practices, planting density, soil pH, weed control, climate conditions, genetic factors and so on.

While she is aware of all the factors that are related to crop yield, the causal relations between these factors and the target variable are either unknown or partially known to her.

Some of these factors are not directly manipulable like rainfall, soil moisture content, soil pH, and soil nitrogen content. However, they can be manipulated using soft interventions such as changing irrigation frequency or adding pesticides. For example, soil moisture would be dependent on both rain and irrigation practices, so even though we cannot directly set soil moisture content to our desired level we can manipulate it by changing irrigation practices.

Adding pesticides could be treated as a soft intervention on both soil nitrogen content and soil pH level. Both these variables might have a nonlinear relation to the amount of pesticide used.

If by some mechanism we could set the soil pH to our desired value that would be equivalent to a hard intervention.

By experimentation, she can manipulate the values of these variables discover the causal relations between them, and consequently design a policy that maximizes crop yield.

## A.9 Simulation Details

### A.9.1 Dropwave

Surjanovic and Bingham [2013], Astudillo and Frazier [2021], Sussex et al. [2022]
For our setting we consider $A_0 \in [-5.12, 5.12]$ and $A_1 \in [-5.12, 5.12]$, we set $\beta = 0.5$ and $\epsilon_i = 0.1 \forall i \in [m]$

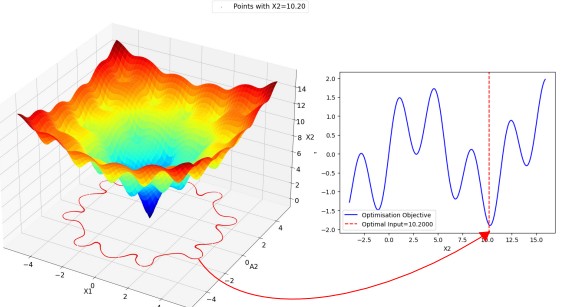

Figure 5: Illustrative example based on Ackley [2012], $X2 = f_2(X1, A2), Y = f_y(X2)$. *Hard Interventions*: X2 can be manipulated therefore optimisation objective can directly be achieved by setting X2 to the correct value. *Soft Interventions* : X2 cannot directly be modified but is a function of the value of X1 and action A2, hence by observing the value of X1 and appropriately choosing A2 (represented by plot on X1, A2 plane) the desired value of X2 can be achieved. This example shows observing intermediate nodes is essential. Consider a modeller trying to optimise $Y$ using $A2$, while ignoring the values of $X1, X2$, they would observe a lot of different values of $y$ for the same action $A2$ considering that $X1$ can take on several different values, consequently affecting X2 and Y.

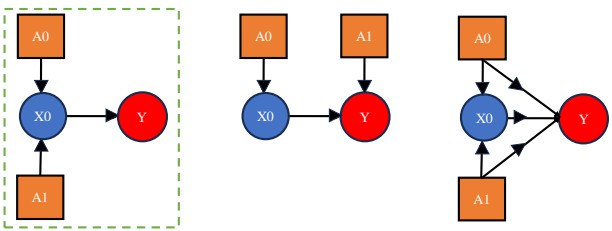

Figure 6: Dropwave: True DAG structure, and Incorrect DAG structures used in Experiment

$$x_0 = f_0(a_0, a_1) = \sqrt{a_0^2 + a_1^2} + \epsilon_0$$
$$y = f_y(x_0) = \frac{1 + \cos(12x_0)}{2 + 0.5x_0^2} + \epsilon_y \tag{18}$$

**Results** GACBO outperforms all cases other than MCBO when the graph is known as apriori. We see a dip in performance in the initial rounds as expected, which are spent on graph discovery, however after the 20th iteration on all seeds the posterior is concentrated around the true graph and the performance matches that of MCBO.

### A.9.2 Rosenbrock

Jamil and Yang [2013], Astudillo and Frazier [2021], Sussex et al. [2022]
For our setting we use $a_i \in [-2, 2]$ for $i \in [m]$, we use $\beta = 0.5$ and $\epsilon_i = 0.1 \forall i \in [m]$

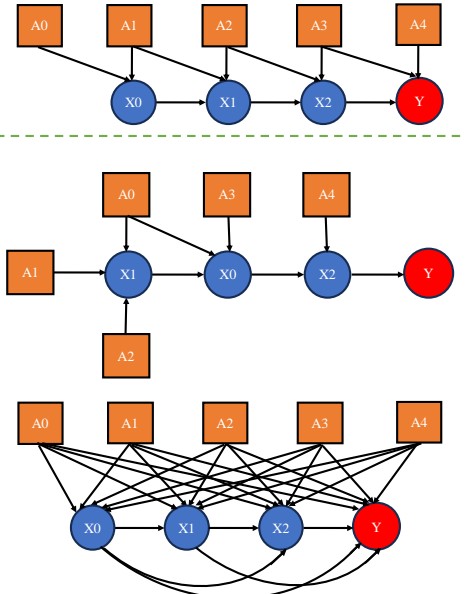

Figure 7: Rosenbrock: True DAG structure, and Incorrect DAG structures used in Experiment

$$f_0(a_0, a_1) = -100a_1 - a_0^2 2 - (1 - a_0)^2 + \epsilon_0$$
$$f_k(a_k, a_{k+1}, x_{k-1}) = -100a_{k+1} - a_k^2 2 - (1 - a_k)^2 + x_{k-1} + \epsilon_k i = 1, \ldots, m \quad (19)$$

**Results** GACBO matches the performance of other baselines in this environment. This behaviour is expected as Rosenbrock has an additive structure and causal graph knowledge does not help accelerate BO's performance. However as demonstrated in our experiments, MCBO with missing edges performs drastically worse as compared to all other methods demonstrating that utilising MCBO with a misspecified graph can result in bad performance.

### A.9.3   Alpine3

[Jamil and Yang, 2013, Sussex et al., 2022, Astudillo and Frazier, 2021] For our experiments we consider $a_i \in [0, 10]$, for $i \in m$, we consider $\beta = 0.5$ and $\eta = 0.1$

$$f_0(x_0) = -\sqrt{x_0} \sin(x_0) + \epsilon_0$$
$$f_i(a_i, x_{i-1}) = \sqrt{a_i} \sin(a_i) x_{i-1} + \epsilon_i, i = 1, \ldots, m \quad (20)$$

**Results** Similar to Dropwave, GACBO outperforms all methods except MCBO with a known graph. For Alpine, the non-linear relationships make knowing the graph structure crucial for faster identification of optimal actions, as shown by the empirical results.

### A.9.4   ToyGraph

Aglietti et al. [2020]

$$X = \epsilon_x$$
$$Z = \exp(-X) + \epsilon_z \quad (21)$$
$$Y = \cos(Z) - \exp(-Z/20) + \epsilon_y$$

**Results** GACBO significantly outperforms MCBO with the wrong graph in both cases (extra and missing edges). The ToyGraph environment is incredibly noisy when intervening on non-parent nodes and therefore knowledge of the graph structure boosts performance.

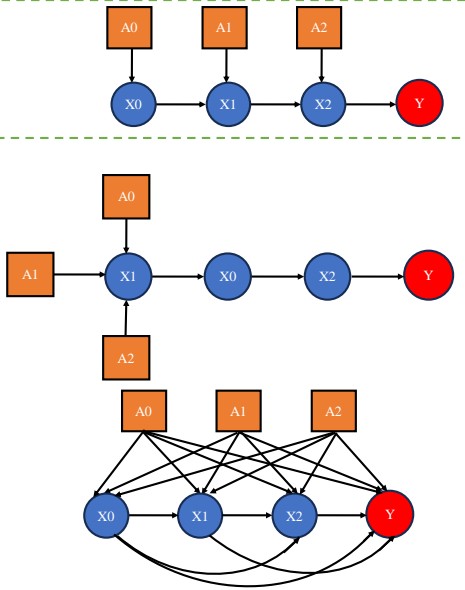

Figure 8: Alpine3: True DAG structure, and Incorrect DAG structures used in Experiment

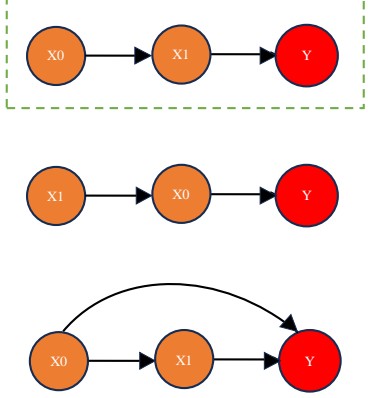

Figure 9: Toy Graph: True DAG structure, and Incorrect DAG structures used in Experiment

### A.9.5 Epidemiology

, Branchini et al. [2023], Havercroft and Didelez [2012]
In our settings, we consider the following input ranges for interventions $T \in [0, 4]$ and $R \in [0, 4]$, we use $\beta = 1$ and noise levels are specified according to the SCM.

$$
\begin{aligned}
B &= \mathcal{U}[-1, 1] \\
T &= \mathcal{U}[4, 8] \\
L &= \text{expit}(0.5T + U) \\
R &= 4 + LT \\
Y &= 0.5 + \cos(4T) + \sin(-L + 2R) + B + \epsilon \text{ with} \epsilon \sim \mathcal{N}(0, 1)
\end{aligned}
\tag{22}
$$

### A.10  Causal Discovery and Causal Bayesian Optimization

Causal discovery from observational data [Verma and Pearl, 2022, Andersson et al., 1997, Spirtes et al., 2000, Chickering, 2002, Friedman and Koller, 2003, Shimizu et al., 2006] can recover causal graphs up to Markov Equivalence Classes (MEC). Friedman and Koller [2003], Janzing et al. [2012] go beyond MEC from purely observational data based on information asymmetry. Tong and Koller [2001], Murphy [2001], Eaton and Murphy [2007], Hauser and Bühlmann [2014], Wang and Jegelka [2017], Ness et al. [2017], Yang et al. [2018], Ghassami et al. [2018], Agrawal et al. [2019], Faria et al. [2022] study the problem of learning graphs from observational and interventional data.

Active causal discovery aims to learn the SCM efficiently. For example, von Kügelgen et al. [2019] studied causal structure learning actively using Bayesian Optimal Experimental Design (BOED). The acquisition function used in their model seeks to select the intervention that is maximally informative about the underlying causal structure with respect to the current model.

Bayesian optimisation aims to learn the optimal point of unknown functions. Knowing the causal structure helps us reduce the causal intrinsic dimension of the optimisation problem for hard intervention. In the case of soft interventions, causal knowledge is useful for utilising the information from intermediate nodes and converting a high-dimension problem into $n$ smaller dimensional optimisation problems (where $n$ is the number of intermediate nodes).

However, learning the entire SCM such as in active causal discovery (i.e., all the causal edges and mechanisms of all nodes in their entire domain) is not necessary for causal Bayesian optimisation. This is understood in two separate cases:

**Hard Intervention**   A hard Intervention mutates the graph, making the intervened node independent of all its parents and ancestors. Lee and Bareinboim [2018] demonstrated that the optimal intervention lies within the parents when there are no unobserved confounders. In such a case learning the causal relation between the ancestors of the parents does not help the underlying goal of causal Bayesian optimisation. Consider the example of ToyGraph, in the true data-generating mechanism $X_1$ is the parent of $Y$, and on performing $\mathrm{do}(X_1 = x_1)$ the value of $Y$ is not affected by the value of $X_0$, hence for optimization knowing the causal direction or mechanism relating $X_0$ and $X_1$ is not required.

**Soft Intervention**   A soft intervention does not mutate the graph, hence learning the causal relations of the ancestral nodes is still relevant to the downstream optimisation problem, however learning the entire causal structure might still be wasteful. If we have determined (specified by expert knowledge or during a certain step of our active causal discovery process) that a certain node is not an ancestor of the target node, then knowing the ancestors or descendants of the node does not contribute to causal Bayesian optimisation. Causal structure in the case of soft intervention utilises values of intermediate nodes to constrain the optimisation problem. Causal structure is only useful when the decomposed problem is simpler than the original problem. For example consider function $f(x_1, x_2) = g(h(x_1), x_2)$, knowing the intermediate value $h(x_1)$ is only useful if the composed function $f$ is more difficult to optimise (because of non-linearity) than the individual functions $g$ and $h$. We observe in our experiments with the Rosenbrock graph in Section 4 that there is no significant advantage of causal structure when the intermediate functions are purely linear.

Our algorithm naturally unifies the two steps of causal discovery and causal Bayesian optimisation by making causal discovery a sub-task of causal Bayesian optimisation. If multiple causal graphs exist within our hypothesis space that explain the data collected up to time step $t$, we only perform an intervention aimed at disambiguation between these graphs if it potentially leads to better rewards than the optimal value observed thus far. Our acquisition function (2) has three maximisations, for a given graph $g \in G_t$ there are several plausible functions for each node $\tilde{f}_{i,t}^g$ and all possible combinations of node functions define the function space for the graph $g$. We use the optimistic reparameterisation trick to find the combination of functions and actions $a_g$ which maximises the target node. We do this for all plausible graphs and compare the best possible value for each graph $g$. We select the plausible graph $g$ with the maximum possible value for the target node and the corresponding action $a_g$ which maximises it. Consider a hypothetical scenario with two different graphs $g_1, g_2$ which disagree on the value of node $i$ for intervention $a$ but the action which maximises the value of target node $y_{g_1}, y_{g_2}$ in $g_1$ and $g_2$ is same $a^*$, and the target node values also agree i.e., $y_{g_1}^* = y_{g_2}^*$ for action $a^*$, even though performing $a$ would help identify the true graph our acquisition function is designed to choose $a^*$. Because $y_{g_1} \leq y_{g_1}^*$ or $y_{g_2} \leq y_{g_2}^*$ for any action $a \neq a^*$.

### A.11 Superexponential Scaling of DAGs and Scalability

The problem setting we addressed in this paper is challenging due to the super-exponential growth of the number of DAGs with the increase in the number of nodes. We only focus on small graphs where all the graphs can be enumerated to study the problem of CBO with unknown graphs in isolation. Our approach can be further improved to be more scalable, by MCMC-based sampling in the space of graphs Giudice et al. [2023], or Differential approaches like DiBS Lorch et al. [2021] in latent spaces or topological ordering of nodes. We leave it as a future work.

Our method in its current state computes the GP score of all possible graph components (all combinations of parents and actions for all observed nodes) and samples graphs based on the GP score and individually optimises and compares all sampled graphs. For larger graphs, the problem becomes intractable as the number of components for which the GP score needs to be calculated increases exponentially. In the initial rounds, the number of graphs that need to be optimised and the number of comparisons that need to be made also increases superexponentially. Causal Bayesian optimisation using the MCBO approach also takes longer for larger graphs.

### A.12 Discussion on Theoretical Analysis

Our approach suggests attaining a similar regret bound to MCBO but with an added constant term. However, our method holds the potential for a superior regret bound by simultaneously exploring the causal structure and exploiting rewards from the outset. Empirical results indicate that our algorithm, GACBO, exhibits a significantly faster increase in average rewards after initial rounds, underscoring its potential for improved regret. The lack of guarantees for the convergence of the posterior to the true graph in finite samples is a major obstacle. A potential theoretical proofing can be achieved by decomposing our regret into two parts:

- Constant term: For the first $n$ samples before learning the true graph, we obtain the constant regret. This is due to the boundness assumption of function (See section 2.3 "Regularity Assumption") $||f_i^{g^*}|| \leq \mathcal{B}_i$. No matter what actions are selected, the upper bound of instant regret can be bounded by $2\mathcal{B}_i$.

- MCBO regret: the second term is the same as the MCBO regret term since after $n$ samples we've discovered the true graph.

**Effect of Graph Knowledge on Optimisation**   Theorem 1 of Sussex et al. [2022] bounds the regret with high probability when the graph is known but functions are unknown in the case of soft intervention as $R_T \leq \mathcal{O}(L_f^N L_\sigma^N \beta_T^N K^N m \sqrt{T \gamma_T})$ where $\gamma_T = \max_i \gamma_{i,T}$, and $N$ denotes the maximum distance from a root node to $V_m$, $K = \max_i |pa_q^*(i)|$ as compared to Standard Bayesian Optimisation that makes no use of graph structure resulting in cumulative regret *exponential* in $m$. Assuming the use of the Squared Exponential Kernel for modelling all functions, $\gamma_T = \mathcal{O}((K + q)(logT)^{K+q+1})$ scales exponentially with respect to $K$ and $q$ the length of each action vector. This results in an expression that scales exponentially in $K, N$. The theorem demonstrates a potentially exponential improvement in the scaling of cumulative regret for possible actions $m \geq K + N$.

For environments allowing hard interventions, the optimisation problem can be reduced to the Causal Intrinsic dimension [Aglietti et al., 2020] if interventions on parents are allowed, Lee and Bareinboim [2018] shows results that the optimal intervention is always found among parents. For environments which do not allow direct interventions on parents, the problem can be studied as a soft intervention for the mutilated graph, by treating the intervened nodes as action nodes and propagating uncertainty through the remaining nodes. For certain graphs, the depth $N$ of the resulting graph can be reduced significantly, consider by figure 1 of Aglietti et al. [2020] with a slight modification where, intervention on the parent nodes $\{X_{100}, Z_{100}\}$ is not possible but we are allowed to intervene on $\{X_{99}, Z_{99}\}$, this allows us to reduce $N = 2$ from $N = 100$, resulting in an exponential improvement in performance.

**Convergence to the True Graph**   As the posterior mass on the graph distribution converges to the dirac delta distribution on the true graph $p(g) \to \delta_{g=g^*}$ the cumulative regret converges to the cumulative regret accrued when the graph is known. For hard interventions the posterior convergence to the true graph is guaranteed under a few assumptions. For soft intervention models DAGs belonging

to Markov Equivalence Classes are further distinguished under the assumptions underpinning the GPN models, considering the functions $f_i$, are not generally invertible [Giudice et al., 2023], the GPN usually suggest higher scores to models admitting the true SEM structure as confirmed in our numerical experiments. For cases where functions are invertible [Hoyer et al., 2008] guarantees identifiability by leveraging the asymmetry of residual noise distributions.

While asymptotic convergence to the true graph structure is guaranteed, there are no known results for finite samples. However, in our numerical experiments, we observe that the graph converges to the essential graph in a small number of samples and potentially observes exponentially less regret as compared to not knowing the graph henceforth. Several studies have considered the problem of learning the causal structure optimally, Murphy [2001], Tong and Koller [2001], Masegosa and Moral [2013], Hauser and Bühlmann [2014], Kocaoglu et al. [2017]. Future work could look at more efficient techniques to learn the structure with finite time guarantees to place an upper bound on the cumulative regret for the case when the graph is unknown apriori.

### A.13 Experiment Details

All our experiments were performed on Google Colab without a GPU or TPU enabled, we used random seeds $47, 42, 73, 66, 13$ for 5 repeats for all given algorithms and given environments.

We use the BoTorch library for both our surrogate models and acquisition functions, specifically implementing the acquisition function with the *qSimpleRegret* module. For each node, we initialize the lengthscales using BoTorch's default prior and set a lower bound on the lengthscale using functionality provided by GPyTorch. Inputs to the individual functions are normalized to the $[0, 1]^d$ range. To fit the hyperparameters, we maximize the marginal log-likelihood.

### A.14 Limitations and Future Work

In the current work we focus on the problem of causal Bayesian optimisation with unknown graphs, however we make several assumptions which may be violated in practice. We assume no unobserved confounders, this assumption is critical to our causal discovery algorithm and our model-based approach for causal Bayesian optimisation is also not resilient to unobserved confounders. We assume additive noise and known noise distribution for each node, this is a strong assumption in practice and needs to be relaxed in future work. Our regularity assumptions might also restrict the application of our method to problems where the relation between a node and its parents is not highly nonlinear. Our current method does not scale well to larger graphs, however, this can be addressed in future work as described in section A.11. We defer providing theoretical guarantees for our method to future work as discussed in A.12.

