# OpenReview forum: "Graph Agnostic Causal Bayesian Optimisation"
_NeurIPS.cc/2024/Workshop/BDU — NeurIPS BDU Workshop 2024 Poster_

### Official Review · Reviewer_psD6 · 2024-09-24
**Good contribution to the CBO literature**

**Rating:** 7
**Confidence:** 3

**Review:**

In this work the authors present an algorithm called Graph Agnostic Causal Bayesian Optimization (GACBO), which performs Causal Bayesian Optimization (CBO) with unknown graphs for the cumulative regret objective.

Section 3, where the authors detail their approach, is mostly written in a very concise and readable manner. A model over the SCM is maintained using what the authors call plausible models and plausible graphs, whose definitions make sense. This model is then used in the acquisition function which searches over the intervention space to maximize the reward variable.
The acquisition function being used makes intuitive sense, but it would be good to know why the authors choose to take the max over the plausible models and graphs instead of averaging over them.
In Equation 2, $\tilde{f}^g_{i,t}$ is introduced but described in Equation 6 - it would be good to relate them.

Results show that GACBO is a promising approach given that it assumes the DAG is unknown. This is especially clear when comparing GACBO to Model-Based Causal Bayesian Optimization (MCBO, which assumes the DAG is known) with the wrong causal graph. Given that the true causal graph is almost always not known, this an important result.

Figures 6,7,8,9 in the Appendix show the wrong graphs which were used for MCBO in the experiments, however the results may be more robust if the causal graphs were learnt using an off-the-shelf search based causal discovery algorithm (which would probably not give the true DAG) before being passed to MCBO.

Section A.11, line 548 suggests the authors are enumerating over the DAG space - in this situation it would be nice to know how $G_t$ changes over time.

Section A.13, line 612 - The choice of seeds raises questions, why were these seeds chosen? Do these seeds result in some assumptions being satsified? That being said, the transparency is good.

Small issues:
- How is $\delta$ chosen?
- Lines 314-319, and 523-528 are a copy-paste except for one paragraph saying Causal Bayesian Optimization and the other saying CBO.
- The paper uses the term mutated graph for the graph after a hard intervention is performed. I believe the exact terminology is mutilated graph, at least in Pearl's 2009 Causality book.
- Line 20: No space before citation.
- Line 27: No space after "However,".
- Line 58: Would be better to move part "... with each node $i \in [m]$ belong to a compact space $\mathcal{ V }_i \subset \mathbb{ R }$" after introducing $Y=V_m$. Also, as far as I am aware, $[m]$ refers to $\{1,\dots,m\}$, not $\{0, \dots, m\}$.
- Line 73: No whitespace between $0$ and $\forall$ - similar issue in Input of Algorithm 1.
- (Below) Line 84: The same term $R_T$ is used for both soft and hard interventions, also missing an equation number.
- Algorithm 1: $a$ is not bold in 3rd line inside for loop.
- Line 105: Missing . after A.6.
- (Below) Line 107, Equation 4: $\forall_i$ instead of $\forall \: i$.
- Line 434: X and Y are not in LaTeX math, also Figure 5 caption.
- Line 481: no space after [...2023]
- Line 513: Bracket not closed.
- Line 521: Reference error.
- Line 579: log

---

### Official Review · Reviewer_L5C9 · 2024-09-24
**Review for "Graph Agnostic Causal Bayesian Optimisation"**

**Rating:** 6
**Confidence:** 4

**Review:**

The paper proposes a methodology for selecting interventions in order to maximize an expected cumulative reward over $T  < \infty $ iterations, where relationships between variables (intervanable and non-intervenable) can be described by a causal graph so that causal effects can be properly defined. The causal graph and the SCM functions are unknown and priors are placed on these.

The paper takes a potentially interesting direction that differs in (again potentially) interesting ways w.r.t. previous work, which is appropriately cited, but several things need to be clarified before it becomes a full, ready paper. In particular some things about how the graph posterior support is defined and the sampling over graphs is done is unclear/potentially wrong, and many formal assumptions are missing.
I will put "Marginally above acceptance threshold" because this is a workshop, but I would not accept the methodology as is for a conference.

Major comments

- In Eq. 2 the posterior over graphs is not defined. There is not a unique way to define posterior over graphs. See for instance Kuipers, J et al 2022 regarding assumptions about parameter modularity etc. Assumptions should be made explicit (faithfulness, causal sufficiency, etc.) as well as their consequences. Also mentioning whether the posterior satisfies any type of Bernstein-von Mises property, or in what sense the posterior converges to the true (unknown) graph. This has consequences for of Eq. 2. that should be elaborated upon. Sample sizes required to learn the posterior over graphs in a suitable sense should be discussed, see e.g. Castelletti, F. et al 2024.
- The definition of the plausible graphs should be justified much more in depth. Also most of this notation in Eq 3: $ \forall_i: \tilde{f}_i^{g} \in \mathcal{H}_k ,  | \tilde{f}_i^{g} |_k \leq \mathcal{B}_i  $ is undefined (am writing k instead of k_i because of rendering issues). Also very unclear why the plausible graphs change over time. How do you know you won't exclude the true graph over time ? Under what assumptions ? Also this means that you have a posterior over graphs with a support that changes over time ? This affects convergence of the posterior to the true graph (in any meaningful sense) for sure.
- You should elaborate on how you propose to calculate (or approximate) the expectations / variances in Eq. 2.
- Why are you * maximizing* over $g \in G_t $ and $\tilde{\boldsymbol{f}}_g \in \mathcal{M}_t^g $  ? You have posterior distributions over these, no ? It is very weird to maximize over the variable if you have a posterior over it ? Why are you not marginalising ? Again also changing the plausble graphs here affects the optimization problem a lot. And how would you maximize over the plausible set ? Nontrivial discrete optimization.
- Is is unclear whether you are proposing a method for selecting both intervention *variables* AND *values* for those variables  as in Aglietti et al 2020 (or only values, which is a substantially easier problem)

Minor comments
- $G_0$ in line 90 still undefined since definition of plausible graphs come later.
- Line 63 should say in a topological order not "topologically"
- Unclear why you consider *both* soft interventions and hard. Do they not require different treatment in your framework ? I may have missed a detail here.
- The sampling algo for graphs, what is the computational complexity (space/time)? Why is it valid ? Why can you avoid graph MCMC as in, e.g., Kuipers and Moffa 2017 for example.

Refs.
- Castelletti, F. and Consonni, G., 2024. Bayesian sample size determination for causal discovery. Statistical Science, 39(2), pp.305-321.
- Kuipers, J. and Moffa, G., 2017. Partition MCMC for inference on acyclic digraphs. Journal of the American Statistical Association, 112(517), pp.282-299.
- Kuipers, J., Suter, P. and Moffa, G., 2022. Efficient sampling and structure learning of Bayesian networks. Journal of Computational and Graphical Statistics, 31(3), pp.639-650.

---

### Decision · Program_Chairs · 2024-10-09

Accept (Poster)